# Effect of a Tabletop Program for Training Emerging Infectious Disease Responses in Nurses at Small- to Medium-Sized Hospitals in Areas with Poor Healthcare Access

**DOI:** 10.3390/healthcare11172370

**Published:** 2023-08-22

**Authors:** Kyung-Sook Cha, Keelyong Lee

**Affiliations:** 1Department of Nursing Science, Sun Moon University, 70 Sunmoon-ro 221 beon-gil, Tangjeong-myeon, Asan-si 31460, Republic of Korea; chamelda@hanmail.net; 2Department of Nursing, Suwon Science College, 288, Seja-ro, Jeongnam-myeon, Hwaseong-si 18516, Republic of Korea

**Keywords:** communicable diseases, infection control, education, healthcare disparities, nurses

## Abstract

This study developed and evaluated a tabletop program for training nurses working in small- to medium-sized hospitals in areas with poor healthcare access for emerging a tabletop program for training emerging infectious disease responses in nurses working at small- to medium-sized hospitals in areas with poor healthcare access and evaluated infectious disease responses. A tabletop program for training responses to emerging infectious diseases was provided to 29 nurses at a general hospital with <300 beds in a region without a tertiary general hospital or healthcare institution with nationally designated inpatient wards for patients with infectious diseases. The 180 min program consisted of an online theoretical lecture (Phase 1), one-on-one drills (Phase 2), and a scenario-based tabletop exercise (Phase 3). To evaluate the program’s effect, pre- and post-knowledge, awareness, and competencies related to responses to emerging infectious diseases were assessed. The mean knowledge score significantly improved from 11.41 ± 2.33 before the training to 16.69 ± 2.48 after the training (z = −4.529, *p* < 0.001). The mean awareness score significantly improved from 80.83 ± 11.94 before the training to 85.45 ± 7.08 after the training (z = −2.335, *p* = 0.020). The mean competence score significantly increased from 67.31 ± 14.75 before the training to 79.38 ± 10.39 after the training (t = −6.187, *p* < 0.001). The tabletop exercise program effectively enhanced the nurses’ competencies in responding to emerging infectious diseases. The training program developed in this study may be utilized in addition to a simplified theoretical lecture to train nurses to improve their competency in responding to emerging infectious diseases.

## 1. Introduction

### 1.1. Rationale of the Research

The continuation of the coronavirus disease (COVID-19) pandemic has adversely affected the field of healthcare worldwide [1]. Healthcare institutions in South Korea have already experienced severe chaos due to the outbreak of Middle East Respiratory Syndrome (MERS) in 2015. Interest in responding to emerging infectious diseases is increasing owing to concerns over the emergence of novel infectious diseases, such as monkeypox, and the influx of infectious diseases, such as avian influenza and the Ebola virus. Early detection of patients with emerging infectious diseases and appropriate healthcare responses contribute significantly to preventing infection spread within healthcare institutions [2] and communities.

In South Korea, patients with emerging infectious diseases are transferred to and treated at hospitals with nationally designated inpatient wards, where the establishment and operation of isolation rooms are funded by the government. Thus, the burden of care for patients with emerging infectious diseases has decreased in general healthcare institutions. However, in Korea, only 38 healthcare institutions operate nationally designated inpatient treatment beds, and there are regional disparities in their distribution. For example, Chungcheongnam-do Province, which is divided into 15 administrative regions (eight cities and seven counties), has only two hospitals that provide healthcare services [3]. Therefore, healthcare institutions in areas with poor healthcare access, where there are no nationally designated inpatient wards, play a crucial role in the prevention of the spread of infectious diseases, as they need to treat and manage infected patients during the delay in the transfer of patients to a hospital with a nationally designated inpatient ward. Hence, the competency of healthcare workers to control infection in small- to medium-sized hospitals in areas with poor healthcare access is critical. However, small- to medium-sized hospitals lack specialists and resources for infection control and have limitations in establishing an adequate emerging infectious disease response system and educating healthcare workers [4]. Accordingly, small- to medium-sized hospitals need special support and training to ensure that healthcare workers are equipped with essential competencies to respond to patients with emerging infectious diseases.

In particular, nurses are at high risk of infection spread because, while providing nursing care, they are in close contact with patients with emerging infectious diseases that they have never encountered previously. Nurses constitute the highest proportion of healthcare workers [5] and play an important role in infection prevention, control, and isolation [6,7]. When infectious diseases emerge, nurses may have difficulty performing nursing tasks during the early phase owing to a lack of information regarding the infection. Therefore, it is crucial to improve education and training to reduce confusion and anxiety in hospitals when emerging infectious diseases are encountered and increase the nurses’ competencies in responding to emerging infectious diseases.

A tabletop exercise is a training approach aimed at improving nurses’ competencies in this regard [8]. A tabletop exercise is a simulation exercise used for training responses to disasters and emergency situations [9] and has been proposed as an effective method for preparing for large-scale bioterror attacks and pandemics [10]. Tabletop exercise participants were instructed to discuss and create realistic and practical response plans according to the characteristics of their own healthcare institution and based on scenarios similar to the settings they are likely to encounter during a real-life outbreak of an infectious disease [11]. Tabletop exercises helped participants increase their knowledge of disasters, clearly recognize their roles and responsibilities, and develop action plans for emergency situations [12]. Ultimately, this training approach can increase the nurses’ confidence [13] by improving their competencies to perform nursing tasks in disaster situations [12,14] and respond appropriately to these situations [9].

Tabletop exercises have been utilized in preparation for the occurrence of emerging infectious diseases [10,11,15,16,17,18]. However, little research has been conducted to examine the effect of tabletop exercises related to emerging infectious diseases administered directly at a healthcare institution [11]. Furthermore, this training approach remains unfamiliar in South Korea.

### 1.2. Study Purposes

The current study aimed to develop a tabletop exercise program for training nurses at small- to medium-sized hospitals in areas with poor healthcare access to respond to emerging infectious diseases and evaluate the program’s effect. The specific objectives of the study were as follows:

First, to develop a tabletop exercise program for nurses in small- to medium-sized hospitals in areas with poor access to healthcare in response to emerging infectious diseases.

Second, to investigate the effect of the program on the nurses’ knowledge of emerging infectious diseases, awareness of measures for responding to emerging infectious diseases, and their competencies in responding to emerging infectious diseases.

### 1.3. Hypotheses

**Hypothesis 1** **(H1).***Program participants’ knowledge of emerging infectious diseases will significantly increase after the implementation of the program*.

**Hypothesis 2** **(H2).***Program participants’ awareness of how to respond to emerging infectious diseases will increase significantly after the implementation of the program*.

**Hypothesis 3** **(H3).***Program participants’ competency scores in responding to emerging infectious diseases will significantly increase after program implementation*.

## 2. Materials and Methods

### 2.1. Study Design

This was a pre-experimental study with a one-group pre-test–post-test design.

### 2.2. Subjects

Convenience sampling was conducted at a single general hospital with fewer than 300 beds (located in S city, C province) in a region without a tertiary general hospital or any healthcare institution with nationally designated inpatient wards for patients with infectious diseases. A flyer was posted at the hospital after obtaining permission to recruit study participants. Nurses who were informed of the study had an understanding of the study purpose and voluntarily consented to participate were selected as the study subjects.

The sample size was estimated using G*power 3.1.2 (https://www.psychologie.hhu.de/arbeitsgruppen/allgemeine-psychologie-und-arbeitspsychologie/gpower, accessed on 1 July 2022). Under the assumptions of a significance level of 0.05, an effect size of 0.50, and a power of 0.80, the minimum sample size required for the paired *t*-tests was *n* = 28. Thirty nurses were recruited, considering the dropout rate. One nurse dropped out of the study; thus, data from 29 participants were analyzed.

### 2.3. Definition of Variables

#### 2.3.1. General Characteristics and Emerging Infectious Diseases-Related Characteristics

General characteristics such as age, sex, job position, and total work experience were examined. Regarding emerging infectious disease-related characteristics, the experience of wearing personal protection equipment (PPE) in response to emerging infectious diseases, receiving training on emerging infectious diseases, level of on-the-job exposure to COVID-19, and on-the-job experience of contact with patients with emerging infectious diseases were included.

#### 2.3.2. Knowledge Regarding Emerging Infectious Diseases

To assess the knowledge regarding emerging infectious diseases, the instrument used by Choi [19] for nurses at healthcare institutions with nationally designated inpatient wards for patients with infectious diseases was revised by current researchers. The revised instrument consists of 19 items on the definition, etiology, transmission mechanisms, symptoms, treatment of emerging infectious diseases, and infection control guidelines. Cronbach’s α value of this instrument was 0.75.

#### 2.3.3. Awareness of Measures for Responding to Emerging Infectious Diseases

The instrument used by Choi [19] to evaluate competency in responding to emerging infectious diseases in nurses at healthcare institutions with nationally designated wards was revised by the current researchers. The revised instrument consisted of 18 items encompassing the early response phase (four items), putting on and taking off PPE (five items), and infection control guidelines (nine items). The items were scored on a 5-point Likert scale, with 1 point for “not at all important” to 5 points for “very important”. The scores range from 18 to 90, with higher scores indicating higher levels of awareness. Cronbach’s α value of this instrument was 0.94.

#### 2.3.4. Competence in Responding to Emerging Infectious Diseases

The instrument, a revision of that proposed by Choi [19], was used to assess competence in responding to emerging infectious diseases. For this evaluation, the revised instrument consisted of 18 items encompassing the early response phase (four items), putting on and taking off PPE (five items), and infection control guidelines (nine items). The items were scored using a 5-point Likert scale, with 1 point for “I cannot perform the task at all” to 5 points for “I can perform the task very well”. The scores range from 18 to 90, with higher scores indicating a higher level of competence. Cronbach’s α value for this instrument was 0.95.

### 2.4. Program Development and Application

Data were collected between 14 November and 25 November 2022. The specific study procedure is as follows.

#### 2.4.1. Development and Test of Tabletop Exercise Program for Emerging Infectious Disease Response

In this study, a tabletop exercise program for emerging infectious disease response was developed to increase nurses’ competency for the accurate assessment of a situation and crisis response when encountering patients with emerging infectious diseases in a hospital ward (such as case reports to the relevant departments, patient isolation and transfer, contact control, and environmental control). The program was designed to provide relevant education and skills training to help nurses understand emerging infectious diseases and respond appropriately when emerging infectious diseases are encountered in inpatients, as well as to teach them how to address problems with the help of a case study.

To develop the program, focus group interviews were conducted online with three infectious disease nurses working in small- to medium-sized hospitals and three infectious disease nurses working in the office of infection control in tertiary general hospitals with a nationally designated inpatient ward. The interview lasted approximately 1 h. Based on the findings of the focus group, MERS, COVID-19, Ebola virus, and monkeypox were identified as emerging infectious diseases likely to be encountered in small- to medium-sized hospitals. Furthermore, how to put on and take off PPE used to stop viral transmission and protect medical staff and how to respond to the occurrence of emerging infectious diseases were identified as topics of lectures and drills. With respect to the training mode, a scenario-based small-group tabletop exercise was developed so that the training would be applicable in practice. To reflect the characteristics of the study institution, the content of the scenario-based tabletop program was revised with the help of a hospital infectious disease nurse.

The program consisted of lectures, drills, and scenario-based tabletop exercises. The details are presented in Table 1. The topics and contents of the lectures and drills were finalized by consulting two nursing professors licensed as infectious disease nurses, five infectious control experts with 15 or more years of experience in infection control, and an infectious disease physician.

#### 2.4.2. Pre- and Post-Test Surveys

The pre-test survey was performed 10 days before the start of the program. The pre-test survey collected data on general and emerging infectious disease-related characteristics and assessed knowledge, awareness, and competence regarding emerging infectious disease responses. The survey took approximately 15–20 min to complete. A post-test on knowledge, awareness, and competence was performed upon completion of the program. Anonymous IDs were assigned to the surveys, and participants were instructed to place the completed surveys in a survey collection box.

#### 2.4.3. Experimental Treatment

The experiment proceeded in a sequence of lectures, followed by drills, and finally, by the scenario-based tabletop exercise (Table 1).

In Phase 1, online lectures were conducted 1 week before the drills. The lecture topics included the status of the occurrence, etiology, transmission routes, symptoms, diagnosis, treatment, and infection prevention and control of MERS, COVID-19, Ebola virus infection, and monkeypox. Each topic was presented using PowerPoint slides incorporating text, graphics, images, and a recording with the instructor’s voice. Approximately 25 min long videos for each topic were uploaded to a personal YouTube channel, allowing individual participants to access and listen to them.

In Phase 2, participants performed drills of putting on and taking off level-D PPE (full-body gowns, N95 masks, goggles, outer and inner gloves, and shoe covers) and performed an N95 mask-fit test in a one-on-one setting. In the PPE drill, participants were instructed to wear and take off the PPE after a nurse from the infection control office who had 5 or more years of experience demonstrated how to put the PPE on and take them off. Participants were provided with immediate feedback and, if necessary, instructed to practice this again. In the N95 mask fit test, cup-shaped masks (8210, 3M, Maplewood, MN, USA) and folded masks (9210+, N95, 3M, Singapore) were used. The mask-fit test was performed using a PORTACOUNT^®^ PRO+8038 Respirator FIT tester (TSI Inc., Shoreview, MN, USA), and test results were explained to participants so that they could select a more appropriate type of mask.

In Phase 3, a scenario-based tabletop exercise was led by an infection control expert using the blueprint of the patient ward of the study institution and a tabletop exercise kit (which included a PPE card, character card, transportation means card, medical device card, environmental control supplies card, and controlled access zone card). The infection control expert worked in the infection control office of a healthcare institution with nationally designated inpatient wards, had more than 15 years of experience, and was a licensed infection control nurse. In the tabletop exercise, a scenario was presented in which a patient with an emerging infectious disease was present in a multi-bed ward. The following questions were discussed, and feedback was provided to encourage the study participants to participate actively in the exercise: 1. What is the very first thing you need to do? 2. To whom do you report the event? 3. How do you transport the patient? 4. Which PPE do you need? 5. What supplies do you require in the isolation room? 6. How extensively do you decide to trace contacts? 7. How do you manage the contacts? 8. How do you handle the medical devices used in the isolation room? 9. How do you handle the supplies used in the isolation room? 10. How do you handle the laundry used by the emerging infectious disease patient? 11. How do you treat the waste generated in the isolation room? and 12. How do you clean and disinfect the isolation room? The exercises were performed in an in-person setting. Three to four participants were assigned to each team so that decision-making would be possible based on lively interactions among team members. The tabletop exercise was conducted for a duration of 60 min per group by an infection control expert.

#### 2.4.4. Ethical Considerations

This study was approved by the Institutional Review Board of Sun Moon University (IRB No. SM-202208-035-2). Prior to data collection, the participants were informed of the study’s purpose and procedure, anonymity, and confidentiality. They were also informed that participation in the study was voluntary and that they could withdraw from the study at any time without any consequences. The participants were given small gifts to show their appreciation for participating in the study. All documents pertaining to the study (including returned surveys) were stored in a locked place in the research’s office, and general and survey data were codified for the researchers’ information only. Finally, participants were informed that the data would be shredded and completely discarded at the end of the 3-year storage period after completion of the study.

### 2.5. Data Analysis

Data analysis was performed using SPSS (version 25.0; IBM SPSS Inc., Armonk, NY, USA), and general and emerging infectious disease-related characteristics were examined by computing frequencies, percentages, means, and standard deviations. Scores for knowledge, awareness, and competence regarding emerging infectious disease responses were examined by computing means and standard deviations. The internal consistency of the study instruments was assessed by computing Cronbach’s α values. Pre–post changes were analyzed using paired t-tests and Wilcoxon signed-rank tests.

## 3. Results

### 3.1. General Characteristics and Emerging Infectious Disease-Related Characteristics

The general and emerging infectious disease-related characteristics of the participants are shown in Table 2.

The mean age of participants was 33.97 ± 9.45, and 93.1% (*n* = 27) of the study sample were female. The proportion of staff nurses was 93.1% (*n* = 27). The mean length of total nursing experience was 73.38 ± 24.30 months, and nurses with less than 1 year of experience comprised 37.9% (n = 11) of the sample and constituted the highest proportion among the groups classified based on the length of nursing experience.

With respect to emerging infectious disease-related characteristics, 75.9% (*n* = 22) received training on how to put on and take off PPE, and 72.4% (*n* = 21) received education on emerging infectious diseases. Regarding on-the-job exposure to COVID-19, 72.4% (n = 21) had direct exposure to patients with a confirmed COVID-19 diagnosis, which is the most common type of exposure to emerging infectious diseases. Regarding on-the-job exposure to patients with (or suspected of having) an emerging infectious disease, 96.6% (*n* = 28) had been exposed to COVID-19, and 3.4% (*n* = 1) had been exposed to patients with novel influenza.

### 3.2. The Program’s Effect on Competence in Responding to Emerging Infectious Diseases

The results of the analyses conducted to test the effect of the tabletop exercise program on responses to emerging infectious diseases are shown in Table 3.

The results supported Hypothesis 1: “Program participants’ knowledge scores regarding emerging infectious diseases will significantly increase from before to after the implementation of the program”. The mean pre-program score for knowledge was 11.41 ± 2.33, whereas the mean post-program score was 16.69 ± 2.48, indicating a significant increase (z = −4.529, *p* < 0.001).

Hypothesis 2, “Program participants’ scores for awareness of how to respond to emerging infectious diseases will significantly increase from before to after implementation of the program”, was also supported. The mean pre-program score of awareness was 80.83 ± 11.94, whereas it significantly improved to 85.45 ± 7.08 after completion of the program (z = −2.335, *p* = 0.020).

Similarly, Hypothesis 3, “Program participants’ scores for competence in responding to emerging infectious diseases will significantly increase from before to after implementation of the program”, was also supported. The mean pre-program score for competence was 67.31 ± 14.75, whereas the mean post-program score was 79.38 ± 10.39, which constituted a significant increase (t = −6.187, *p* < 0.001).

## 4. Discussion

This study was conducted to develop a tabletop exercise program for training nurses in small- to medium-sized hospitals in areas with poor healthcare access to respond to emerging infectious diseases and to test the effect of the program by measuring changes in nurses’ knowledge, awareness, and competence regarding emerging infectious disease responses.

The nurses who participated in the tabletop exercise program developed in the study showed a mean knowledge score that was significantly higher than that prior to participation in the program. This confirmed that the program effectively improved knowledge of emerging infectious diseases. No previous studies conducted with nurses have measured changes in knowledge regarding emerging infectious diseases after participating in hospital-based programs for training responses to emerging infectious diseases, making direct comparisons impossible. Nevertheless, the current study’s findings were in line with those of a previous study that used a tabletop exercise in response to severe acute respiratory syndrome (SARS), wherein public health workers showed increased knowledge of SARS after completing the program [13]. Park [20] mentioned the need for a training approach in which trainees can experience disaster situations to improve their knowledge. For more efficient delivery of knowledge, the program used in this study was designed such that participants performed a scenario-based tabletop exercise in a small-group setting after receiving relevant lectures and conducting drills. We speculate that this design will increase the efficiency of training programs. In a study that compared the effect of disaster response training based on lectures vs. tabletop exercises, Mirzaei [21] reported that although knowledge scores improved after training in both groups, the group that was trained using the tabletop exercise, as well as the lecture, scored higher than the group that received the lecture only. Thus, an approach based on both lectures and tabletop exercises enhances the efficiency of education/training of responses to emerging infectious diseases and should be considered when designing training programs.

The mean awareness score significantly increased after the training program. This demonstrates that the tabletop exercise program for emerging infectious disease responses positively impacted the nurses’ awareness. Few intervention studies have examined the awareness of emerging infectious disease responses; thus, intervention studies should continue to be performed. Awareness of the seriousness of emerging infectious diseases may be useful for predicting the attitude and behavior of emergency room nurses in the event of emerging infectious disease encounters [22]. Yi and Cha [23] stated that, in general, those with high awareness may go beyond the guidelines for performing infection control tasks, whereas those with low awareness may fail to follow these guidelines. Through training in emerging infectious disease responses, nurses may better recognize their roles, which may steer them to respond appropriately to the occurrence of emerging infectious diseases.

The mean post-program competence score was also significantly higher than the pre-program score, confirming that the tabletop exercise program for emerging infectious disease response effectively improved nurses’ competencies. No previous studies have used the same program, participants, or research instruments, making direct comparisons difficult. However, our study findings were consistent with previous findings that public health workers’ skills pertaining to SARS were enhanced after they had received tabletop exercise-based training for responding to SARS [15] and that nurses’ disaster nursing competencies were improved after they had received tabletop exercise-based training for disaster responses [24,25]. Nursing competencies in disaster situations (including emerging infectious diseases) require competencies pertaining to patient severity classification, PPE, and psychological nursing care skills. Hence, theory-focused education should be avoided, and training that includes drills should be used [26]. The tabletop exercise in the current study was performed using the blueprint of the study institution’s ward based on a scenario, and a discussion format was used. Participants were asked questions regarding the situation they might encounter with patients with emerging infectious diseases in the ward where they were working and were encouraged to find measures to respond to the situation by discussing it with team members. In this process, participants developed realistic and practical response plans appropriate to the characteristics of their hospital. It is speculated that training that can be immediately applied to their jobs positively impacts confidence and competence in responding to emerging infectious diseases. Thornton [10] emphasized the value of tabletop exercises involving response staff and highlighted that their effectiveness depends on proper design, careful execution, thorough evaluation, and actual implementation by the participating entity. Therefore, to increase the effectiveness of emerging infectious disease response training, programs should be designed to include drills applicable to the actual situation and reflect the characteristics of the participants’ workplace such that the training can be directly applied when they perform their tasks. Furthermore, it is important to continuously adapt training to the evolving work environments, available resources, and personnel conditions. Regularly evaluating the training outcomes should be made to ensure that it remains relevant and effective in preparing response staff for challenging situations. The operational capabilities of the response staff can be significantly improved by consistently integrating changes and evaluating the training results.

According to a previous study [23], the performance of infection control for emerging infectious diseases was better in nurses who received two or more training sessions on emerging infectious diseases and PPE than in nurses who did not receive such training and in nurses who received repeated training during an outbreak than in nurses who did not. Mirzaei [21] found that the level of nurses’ knowledge regarding disaster responses had decreased by 4 weeks after as compared to immediately after a tabletop exercise. Kim [24] reported that competency in disaster nursing significantly increased immediately after a tabletop exercise but showed a decreasing trend after 4 weeks. In contrast, Aliakbari [27] confirmed that the disaster response capabilities of nurses, including tabletop exercises, remained even 3 months after the intervention. Given the unpredictable nature of emerging infectious diseases in terms of timing and characteristics, it is necessary to provide regular and repetitive training on response measures to enhance the practical skills of the response staff. The timing of repeated training should be determined by considering factors such as the duration of training effectiveness and other relevant considerations.

## 5. Conclusions

To the best of our knowledge, this study is the first to evaluate the effect of a scenario-based tabletop exercise program targeting nurses at small- to medium-sized hospitals in areas with poor healthcare access amidst the global pandemic of COVID-19. The tabletop exercise program effectively improved nurses’ knowledge, awareness, and competence regarding emerging infectious diseases. We suggest that the education/training program developed in this study be utilized if training is planned to enhance nurses’ competencies in responding to emerging infectious diseases.

A limitation of this study is that the tabletop exercise program for emerging infectious disease response was administered to a single group, thereby limiting the comprehensive evaluation of the program’s overall effectiveness. It was difficult to select study institutions because of the prolonged COVID-19 pandemic and South Korea’s continuing and thorough COVID-19 policies, as external infection control experts required access to healthcare facilities to conduct in-person training sessions with nurses. Consequently, the research team was unable to find approved medical institutions, resulting in the inability to establish a control group.

Future research should include experimental or multi-institutional studies to compare and analyze the effectiveness of the tabletop exercise programs in experimental and control groups. Additionally, tabletop exercises targeting nurses should be expanded, and a training program should be developed to target various healthcare workers to improve overall emerging infectious disease responses.

## Figures and Tables

**Table 1 healthcare-11-02370-t001:** Tabletop exercise program for training emerging infectious disease responses.

Phase	Class Times (min)	Topics	Content	Method
1 (Lecture)	25	Understanding emerging infectious diseases 1 (MERS, COVID-19)	- The status of occurrences, etiology, transmission routes, symptoms, diagnosis, treatment, and infection prevention and control	Online lecture
	25	Understanding emerging infectious diseases 2 (Ebola virus infection, monkeypox)	- The status of occurrence, etiology, transmission routes, diagnosis, treatment, and infection prevention and control	Online lecture
	20	PPE	- Putting on and taking off level-D PPE (full-body gowns, N95 masks, goggles, outer and inner gloves, shoe covers) - Putting on and taking off five PPE types (long disposable gowns, N95 masks, goggles, and outer and inner gloves)	Online lecture
	20	Nurses’ response in the occurrence of emerging infectious disease patients	- Nurses’ response (patient occurrence report, patient assignment, request for an epidemiological survey, patient management, patient transport, environmental control after patient transfer to another room, and contact management)	Online lecture
2 (Drill)	30	Putting on and taking off PPE	- Putting on and taking off level-D PPE - N95 mask-fit test	1:1 exercise
3 (Tabletop Exercise)	60	Scenario-based tabletop exercise	- Scenario (a patient with an emerging infectious disease occurred in a multi-bed ward was presented) - Nurses’ response plan (patient occurrence report, interdepartmental communication, patient transport route verification, PPE selection, isolation room preparedness, isolation-related item management, medical devices management, environmental management, and close contacts management) - Expert feedback	Group exercise (3–4 persons/group), Group discussion

**Table 2 healthcare-11-02370-t002:** General characteristics and emerging infectious disease-related characteristics (*n* = 29).

	Variable	Categories	N (%) or M ± SD (Range)
General characteristics	Age (years)	≤29 30–39 ≥40	12 (41.4) 5 (17.2) 12 (41.4) 33.97 ± 9.45 (23–49)
	Sex	Male Female	2 (6.9) 27 (93.1)
	Job position	Head nurse Staff nurse	2 (6.9) 27 (93.1)
	Total nursing experience (months)	≤11 12–36 37–119 ≥120	11 (37.9) 5 (17.3) 4 (13.8) 9 (31.0) 73.38 ± 24.30 (1–252)
Emerging infectious disease-related characteristics	Received training on PPE	Yes No	22 (75.9) 7 (24.1)
	Received education on emerging infectious diseases	Yes No	21 (72.4) 8 (27.6)
	Level of on-the-job exposure to COVID-19 *	Contact with a confirmed patient. Proximity within 2 ms Having been in the same indoor space Specimen handling (e.g., test swab, sputum) Contact with a patient suspected of COVID-19	21 (72.4) 3 (10.3) 5 (17.2) 7 (24.1) 5 (17.2)
	Experience in contacting emerging infectious diseases on the job *	COVID-19 Novel influenza	28 (96.6) 1 (3.4)

* Multiple responses are allowed.

**Table 3 healthcare-11-02370-t003:** Effects of the tabletop exercise program for emerging infectious disease response (*n* = 29).

Variables	Categories	Pre-Test (M ± SD)	Post-Test (M ± SD)	t or z	*p*-Value
Knowledge regarding emerging infectious diseases	Definition	0.79 ± 0.41	0.83 ± 0.38	−0.302 *	0.763
	Etiology	0.21 ± 0.41	0.69 ± 0.47	−3.500 *	<0.001
	Mechanisms of transmission	2.69 ± 1.22	4.45 ± 0.73	−4.249 *	<0.001
	Symptoms	2.10 ± 0.81	3.55 ± 0.68	−4.477 *	<0.001
	Treatment	0.48 ± 0.50	0.90 ± 0.31	−3.464 *	<0.001
	Infection control guidelines	5.03 ± 0.86	6.28 ± 0.84	−4.036 *	<0.001
	Total	11.41 ± −2.33	16.69 ± 2.48	−4.529 *	<0.001
Awareness of the measures for responding to emerging infectious diseases	Early response phase	17.62 ± 2.89	19.14 ± 1.45	−2.898 *	0.004
	Putting on and taking off PPE	22.83 ± 3.29	23.69 ± 2.03	−1.473 *	0.141
	Infection control guidelines	40.38 ± 6.17	42.62 ± 4.01	−1.995 *	0.046
	Total	80.83 ± 11.94	85.45 ± 7.08	−2.335 *	0.020
Competence in responding to emerging infectious diseases	Early response phase	14.31 ± 2.98	17.38 ± 2.44	−6.557	<0.001
	Putting on and taking off PPE	18.72 ± 4.47	22.17 ± 2.87	−5.466	<0.001
	Infection control guidelines	34.28 ± 7.95	39.83 ± 5.67	−4.912	<0.001
	Total	67.31 ± 14.75	79.38 ± 10.39	−6.187	<0.001

* Wilcoxon signed-rank test.

## Data Availability

The data obtained are available in an anonymized format so that individuals cannot be identified. The data were requested by the authors.

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
