# Peer review of "Effect of a Tabletop Program for Training Emerging Infectious Disease Responses in Nurses at Small- to Medium-Sized Hospitals in Areas with Poor Healthcare Access"

_healthcare, 2023, doi:10.3390/healthcare11172370_

Round 1

Reviewer 1 Report

The undertaken research problem is extremely important for medical personnel. Nursing assistance is essential for the patient and each innovation
is a valuable factor supporting her work. We await further publications by the authors with interest.

Please check thoroughly punctuation, collocations and articles. Please check thoroughly punctuation, collocations and articles.

Author Response

I sincerely appreciate your evaluation, esteemed reviewer.

Reviewer 2 Report

It has been a useful study. But the explanation about how to teach is not complete. Is the educational content subject to individual or group approval? Although it would have been better if there were two groups and the results of the control group and the intervention group were compared before and after the training.

The necessity of studying in deprived areas is not well explained.

Author Response

Point 1:  It has been a useful study. But the explanation about how to teach is not complete.

Response 1:  In order to enhance the reproducibility of the study, detailed information on each phase (Phase 1, Phase 2, and Phase 3) was added in the research methodology. Furthermore, revisions were made to Table 1 to improve the clarity of the study.

“In Phase 1, online lectures were conducted 1 week before the drills. The lecture topics included the status of the occurrence, etiology, transmission routes, symptoms, diagnosis, treatment, and infection prevention and control of MERS, COVID-19, Ebola virus infection, and monkeypox. Each topic was presented using PowerPoint slides incorporating text, graphics, images, and a recording with the instructor’s voice. Approximately 25-min long videos for each topic were uploaded to a personal YouTube channel, allowing individual participants to access and listen to them.

In Phase 2, participants performed drills of putting on and taking off level-D PPEs (full-body gowns, N95 masks, goggles, outer and inner gloves, and shoe covers) and performed an N95 mask-fit test in a one-on-one setting. (ellipsis)

In Phase 3, a scenario-based tabletop exercise was led by an infection control expert using the blueprint of the patient ward of the study institution and a tabletop exercise kit (which included a PPE card, character card, transportation means card, medical device card, environmental control supplies card, and controlled access zone card). (ellipsis) The tabletop exercise was conducted for a duration of 60 min per group by an infection control expert.”

Point 2:  Is the educational content subject to individual or group approval? Although it would have been better if there were two groups and the results of the control group, and the intervention group were compared before and after the training.

 Response 2: The present study obtained confirmation and approval for the educational content through the Institutional Review Board (IRB). Due to the prolonged COVID-19 pandemic, we encountered considerable difficulties in selecting research institutions and thus, were unable to establish a control group for the study. This limitation has been added to the discussion of the research.

“A limitation of this study is that the tabletop exercise program for emerging infectious disease response was administered to a single group, thereby limiting the comprehensive evaluation of the program's overall effectiveness. It was difficult to select study institutions because of the prolonged COVID-19 pandemic and South Korea’s continuing and thorough COVID-19 policies, as external infection control experts required access to healthcare facilities to conduct in-person training sessions with nurses. Consequently, the research team was unable to find approved medical institutions, resulting in the inability to establish a control group. Future research should include experimental or multi-institutional studies to compare and analyze the effectiveness of the tabletop exercise programs in experimental and control groups.”

Point 3: The necessity of studying in deprived areas is not well explained.

Response 3: In the introduction, additional information has been provided to emphasize the importance of managing emerging infectious diseases in the specific area where the research is conducted.

“Thus, the burden of care for patients with emerging infectious diseases has decreased in general healthcare institutions. However, in Korea, only 38 healthcare institutions operate nationally designated inpatient treatment beds, and there are regional disparities in their distribution. For example, Chungcheongnam-do Province, which is divided into 15 administrative regions (eight cities and seven counties), has only two hospitals that provide healthcare services [3].”

I sincerely appreciate your evaluation, esteemed reviewer.

Reviewer 3 Report

In my opinion this issue is very important in onnection with last pandemic COVID-19 and not only.
Introduction is celar, provide sufficient background and include all relevant references. Research design is appropriate and methods adequately described. Results is to little in my opinion, you sholud add some table or picture. Maybe you sholud better described nurses knowledge before and after research? Please add more information about program Tabletop. Conclusions should be supported by the results more. Maybe you can describet the same or similar programs in another country? Cited references is ok, relevant to the research. Need little corrections.

Author Response

Point 1:  Results is too little in my opinion, you should add some table or picture. Maybe you should better describe nurses’ knowledge before and after research?

 Response 1:  In the research results, additional information was provided by analyzing the Effects of the tabletop exercise program for emerging infectious disease response in Table 3, examining each specific item in detail. This was done to offer more comprehensive insights and a deeper understanding of the outcomes of the tabletop exercise program in responding to emerging infectious diseases.

Point 2: Please add more information about program Tabletop.

Response 2:  In order to enhance the reproducibility of the study, detailed information on each phase (Phase 1, Phase 2, and Phase 3) was added in the research methodology. Furthermore, revisions were made to Table 1 to improve the clarity of the study.

“In Phase 1, online lectures were conducted 1 week before the drills. The lecture topics included the status of the occurrence, etiology, transmission routes, symptoms, diagnosis, treatment, and infection prevention and control of MERS, COVID-19, Ebola virus infection, and monkeypox. Each topic was presented using PowerPoint slides incorporating text, graphics, images, and a recording with the instructor’s voice. Approximately 25-min long videos for each topic were uploaded to a personal YouTube channel, allowing individual participants to access and listen to them.

In Phase 2, participants performed drills of putting on and taking off level-D PPEs (full-body gowns, N95 masks, goggles, outer and inner gloves, and shoe covers) and performed an N95 mask-fit test in a one-on-one setting.

Point 3: Conclusions should be supported by the results more. Maybe you can describe the same or similar programs in another country?

Response 3:  In the discussion, the following content was added to enrich the discussion and provide a more profound insight into the implications of the research.

“Thornton [10] emphasized the value of tabletop exercises involving response staff and highlighted that their effectiveness depends on proper design, careful execution, thorough evaluation, and actual implementation by the participating entity. (ellipsis)

In contrast, Aliakbari [27] confirmed that the disaster response capabilities of nurses, including tabletop exercises, remained even 3 months after the intervention. Given the unpredictable nature of emerging infectious diseases in terms of timing and characteristics, it is necessary to provide regular and repetitive training on response measures to enhance the practical skills of the response staff. The timing of repeated training should be determined by considering factors such as the duration of training effectiveness and other relevant considerations.”

I sincerely appreciate your evaluation, esteemed reviewer.

Reviewer 4 Report

The study is very interesting on the training of nurses in emerging infectious diseases, but in my opinion, it needs some minor modifications.

Introduction. It would be interesting to include in Rationale of the research some data on the relevance of emerging infectious diseases in the area where the study takes place. 

The methodology seems to me to be adequate

The results are well presented and clear, although they are too brief.

The discussion is correct.

The conclusions include aspects that should be included in a "limitations" and "future research" section at the end of the discussion.

Author Response

Point 1:  Introduction. It would be interesting to include in Rationale of the research some data on the relevance of emerging infectious diseases in the area where the study takes place. 

Response 1:  In the introduction, additional information has been provided to emphasize the importance of managing emerging infectious diseases in the specific area where the research is conducted.

“Thus, the burden of care for patients with emerging infectious diseases has decreased in general healthcare institutions. However, in Korea, only 38 healthcare institutions operate nationally designated inpatient treatment beds, and there are regional disparities in their distribution. For example, Chungcheongnam-do Province, which is divided into 15 administrative regions (eight cities and seven counties), has only two hospitals that provide healthcare services [3].”

 Point 2:  The results are well presented and clear, although they are too brief.

 Response 2:  In the research results, additional information was provided by analyzing the Effects of the tabletop exercise program for emerging infectious disease response in Table 3, examining each specific item in detail. This was done to offer more comprehensive insights and a deeper understanding of the outcomes of the tabletop exercise program in responding to emerging infectious diseases.

Point 3: The conclusions include aspects that should be included in a “limitation” and "future research" section at the end of the discussion.

Response 3:  In the conclusion, potential directions for future research and the limitations of the study were presented in a separate section.

“A limitation of this study is that the tabletop exercise program for emerging infectious disease response was administered to a single group, thereby limiting the comprehensive evaluation of the program's overall effectiveness. It was difficult to select study institutions because of the prolonged COVID-19 pandemic and South Korea’s continuing and thorough COVID-19 policies, as external infection control experts required access to healthcare facilities to conduct in-person training sessions with nurses. Consequently, the research team was unable to find approved medical institutions, resulting in the inability to establish a control group.

Future research should include experimental or multi-institutional studies to compare and analyze the effectiveness of the tabletop exercise programs in experimental and control groups.”

I sincerely appreciate your evaluation, esteemed reviewer.

Reviewer 5 Report

The study examining the training of nurses in the context of emerging infectious diseases is highly fascinating. However, in my opinion, there are a few areas where minor modifications could enhance its overall impact and effectiveness.

Regarding the introduction, it would be a bit more important to include additional information within the research that emphasizes the relevance and prevalence of emerging infectious diseases in the specific area where the study is conducted. By providing data that highlights the local context and the significance of these diseases, the study can establish a stronger foundation for its objectives and justify the need for effective nurse training.

Moving on to the methodology, it appears to be well-designed and suitable for achieving the study's objectives. The approach taken in training the nurses seems appropriate and aligned with the purpose of addressing emerging infectious diseases. However, providing more explicit details about the specific techniques, tools, or resources used in the training process could enhance the clarity and replicability of the study.

When it comes to the presentation of results, they are commendably well-organized and presented in a clear manner. However, they appear to be rather concise and may benefit from being more comprehensive. By providing additional information, such as specific findings or key statistics, the results section can offer a more thorough understanding of the outcomes and their implications.

Moving forward to the discussion, it seems to adequately address the study's findings and their implications. The analysis and interpretation provided offer a meaningful exploration of the results. However, expanding on certain points or considering alternative explanations could further enrich the discussion and provide a deeper insight into the implications of the study.

Lastly, the conclusions drawn in the study are relevant and appropriately summarizing the main findings. However, it would be beneficial to include certain aspects as part of a separate section dedicated to limitations and future research. By explicitly addressing the study's limitations and suggesting potential avenues for future investigation, the conclusions can become more comprehensive and provide valuable guidance for further exploration in the field.

Overall, the study on training nurses in emerging infectious diseases is engaging and holds considerable potential. By incorporating the suggested modifications, it can strengthen its foundation, provide a more detailed account of the results, and offer a comprehensive roadmap for future research endeavors.

well written

Author Response

Point 1:  Regarding the introduction, it would be a bit more important to include additional information within the research that emphasizes the relevance and prevalence of emerging infectious diseases in the specific area where the study is conducted. By providing data that highlights the local context and the significance of these diseases, the study can establish a stronger foundation for its objectives and justify the need for effective nurse training.

Response 1:  In the introduction, additional information has been provided to emphasize the importance of managing emerging infectious diseases in the specific area where the research is conducted.

“Thus, the burden of care for patients with emerging infectious diseases has decreased in general healthcare institutions. However, in Korea, only 38 healthcare institutions operate nationally designated inpatient treatment beds, and there are regional disparities in their distribution. For example, Chungcheongnam-do Province, which is divided into 15 administrative regions (eight cities and seven counties), has only two hospitals that provide healthcare services [3].”

 Point 2:  Moving on to the methodology, it appears to be well-designed and suitable for achieving the study's objectives. The approach taken in training the nurses seems appropriate and aligned with the purpose of addressing emerging infectious diseases. However, providing more explicit details about the specific techniques, tools, or resources used in the training process could enhance the clarity and replicability of the study.

 Response 2:  In order to enhance the reproducibility of the study, detailed information on each phase (Phase 1, Phase 2, and Phase 3) was added in the research methodology. Furthermore, revisions were made to Table 1 to improve the clarity of the study.

“In Phase 1, online lectures were conducted 1 week before the drills. The lecture topics included the status of the occurrence, etiology, transmission routes, symptoms, diagnosis, treatment, and infection prevention and control of MERS, COVID-19, Ebola virus infection, and monkeypox. Each topic was presented using PowerPoint slides incorporating text, graphics, images, and a recording with the instructor’s voice. Approximately 25-min long videos for each topic were uploaded to a personal YouTube channel, allowing individual participants to access and listen to them.

In Phase 2, participants performed drills of putting on and taking off level-D PPEs (full-body gowns, N95 masks, goggles, outer and inner gloves, and shoe covers) and performed an N95 mask-fit test in a one-on-one setting.

(ellipsis)

In Phase 3, a scenario-based tabletop exercise was led by an infection control expert using the blueprint of the patient ward of the study institution and a tabletop exercise kit (which included a PPE card, character card, transportation means card, medical device card, environmental control supplies card, and controlled access zone card). (ellipsis) The tabletop exercise was conducted for a duration of 60 min per group by an infection control expert.”

Point 3:  When it comes to the presentation of results, they are commendably well-organized and presented in a clear manner. However, they appear to be rather concise and may benefit from being more comprehensive. By providing additional information, such as specific findings or key statistics, the results section can offer a more thorough understanding of the outcomes and their implications.

Response 3:  In the research results, additional information was provided by analyzing the Effects of the tabletop exercise program for emerging infectious disease response in Table 3, examining each specific item in detail. This was done to offer more comprehensive insights and a deeper understanding of the outcomes of the tabletop exercise program in responding to emerging infectious diseases.

Point 4: Moving forward to the discussion, it seems to adequately address the study's findings and their implications. The analysis and interpretation provided offer a meaningful exploration of the results. However, expanding on certain points or considering alternative explanations could further enrich the discussion and provide a deeper insight into the implications of the study.

Response 4: In the discussion, the following content was added to enrich the discussion and provide a more profound insight into the implications of the research.

“Thornton [10] emphasized the value of tabletop exercises involving response staff and highlighted that their effectiveness depends on proper design, careful execution, thorough evaluation, and actual implementation by the participating entity. (ellipsis)

In contrast, Aliakbari [27] confirmed that the disaster response capabilities of nurses, including tabletop exercises, remained even 3 months after the intervention. Given the unpredictable nature of emerging infectious diseases in terms of timing and characteristics, it is necessary to provide regular and repetitive training on response measures to enhance the practical skills of the response staff. The timing of repeated training should be determined by considering factors such as the duration of training effectiveness and other relevant considerations.”

Point 5: Lastly, the conclusions drawn in the study are relevant and appropriately summarizing the main findings. However, it would be beneficial to include certain aspects as part of a separate section dedicated to limitations and future research. By explicitly addressing the study's limitations and suggesting potential avenues for future investigation, the conclusions can become more comprehensive and provide valuable guidance for further exploration in the field.

 Response 5:  In the conclusion, potential directions for future research and the limitations of the study were presented in a separate section.

“A limitation of this study is that the tabletop exercise program for emerging infectious disease response was administered to a single group, thereby limiting the comprehensive evaluation of the program's overall effectiveness. It was difficult to select study institutions because of the prolonged COVID-19 pandemic and South Korea’s continuing and thorough COVID-19 policies, as external infection control experts required access to healthcare facilities to conduct in-person training sessions with nurses. Consequently, the research team was unable to find approved medical institutions, resulting in the inability to establish a control group.

Future research should include experimental or multi-institutional studies to compare and analyze the effectiveness of the tabletop exercise programs in experimental and control groups.”

I sincerely appreciate your evaluation, esteemed reviewer.
